# The Influence of the Genotype and Planting Density on the Structure and Composition of Root and Rhizosphere Microbial Communities in Maize

**DOI:** 10.3390/microorganisms11102443

**Published:** 2023-09-28

**Authors:** Yang Zhang, Jianxin Lin, Shanhu Chen, Heding Lu, Changjian Liao

**Affiliations:** Technical Research Center of Dry Crop Variety Breeding in Fujian Province, Institute of Crops Research, Fujian Academy of Agricultural Sciences, Fuzhou 350013, China; zywheat@163.com (Y.Z.); linjx0522@163.com (J.L.); liaocj1978@163.com (S.C.); luhd137@163.com (H.L.)

**Keywords:** Maize, microbial communities, high-density planting, rhizosphere bacteria, root-associated bacteria

## Abstract

Maize has the largest cultivation area of any crop in the world and plays an important role in ensuring food security. High-density planting is essential for maintaining high maize yields in modern intensive agriculture. Nonetheless, how high-density planting and the tolerance of individual genotypes to such planting shape the root-associated microbiome of maize is still unknown. In this study, we analyzed the root and rhizosphere bacterial communities of two maize accessions with contrasting shoot architectures grown under high- and low-density planting conditions. Our results suggested that maize hosted specific, distinct bacterial communities in the root endocompartment and that the maize genotype had a significant effect on the selection of specific microbes from the rhizosphere. High-density planting also had significant effects on root-associated bacterial communities. Specifically, genotype and high-density planting coordinated to shape the structure, composition, and function of root and rhizosphere bacterial communities. Taken together, our results provide insights into how aboveground plant architecture and density may alter the belowground bacterial community in root-associated compartments of maize.

## 1. Introduction

High-density cultivation is necessary for obtaining high grain yields from a limited field area [1]. Maize has the largest cultivation area of any crop in the world and plays an important role in ensuring food security [2]. However, given the increasing global population and the decline in maize planting areas, there is a need for ever-increasing maize yields, and increasing the yield per unit area is a major strategy for solving this problem [3,4,5].

Light has a significant effect on plant survival and growth through its influence on photosynthetic rates and plant morphology [6]. Maize is a C_4_ crop with high light efficiency, strong density tolerance, and high yield potential. Dense planting increases dry matter accumulation by increasing the number of ears per unit land area, but it reduces the yield per plant, although this can be overcome by field management [7]. The biggest problem associated with dense planting is the reduction in photosynthetic efficiency [8]. Therefore, breeding maize for high-density planting and increasing its photosynthetic efficiency are the major approaches to increasing yield per unit area [9].

Unlike animals, plants are sessile and must therefore overcome various biotic and abiotic environmental challenges to optimize their growth and development [10]. Plant roots have been colonized by complex multi-kingdom microbial consortia since the earliest establishment of ancestral plant lineages on land [11]. The plant root microbiota typically includes archaea, fungi, bacteria, and protists [12,13]. Associations between plants and beneficial microbes have been shown to help plants deal with environmental stress [14,15,16].

Roots are the major sites of plant association with soil microbes [17]: host plants offer carbohydrates derived from photosynthesis to the root-associated microbes, and soil microbes perform multiple beneficial functions for the host plants [18,19,20]. Root exudates are involved in the recruitment and enrichment of root-associated microbes [21]. Nonetheless, little is known about how maize planting density influences root and rhizosphere microbiota. Here, we studied two fresh-eating maize accessions that are widely cultivated in Fujian Province and differ in their shoot architecture and tolerance of high-density planting. We investigated the influence of the maize genotype, planting density, and their interaction on the composition and function of root and rhizosphere bacterial communities. Our results provide insight into the influence of the maize genotype and planting density on the interaction between soil microbes and maize roots.

## 2. Materials and Methods

### 2.1. Maize Accessions and Experimental Field Site

The fresh-eating maize accessions Guangliangtian27 (L27), which was selected and bred by Guangdong Seed Introduction Service Company, and Mintian6855 (L6855), which was selected by the Crop Research Institute of Fujian Academy of Agricultural Sciences, were used in this study. These two accessions were chosen because they are widely cultivated in Fujian Province and exhibit distinctive shoot architecture traits. G27 has a broader shoot architecture and is intolerant of high-density planting, whereas M6855 has a compact shoot architecture and tolerates high-density planting (Appendix A). The experimental site was located at the Field Experimental Station of Fujian Academy of Agricultural Sciences, Fuqing, Fuzhou, Fujian province. The basic soil properties are as follows: soil pH is 5.9, organic matter is 20 g/kg, available N is 100.7 mg/kg, available *p* is 3.4 mg/kg, available *k* is 131.7 mg/kg, total *n* is 0.118%, total *p* is 0.033%, total *k* is 1.38%.

### 2.2. Planting Density Design

For the planting density treatments, maize plants of each accession were cultivated in separate, replicated plots. In the low-density treatment, 96 maize plants were cultivated in plots 500 cm in length and 480 cm in width with a row spacing of 60 cm and an interplant distance of 36 cm. For the high-density treatment, 144 maize plants were cultivated in plots of the same size with a row spacing of 60 cm and an interplant distance of 24 cm (Appendix A). There were five plots for each accession and density level for a total of 20 plots arranged in a randomized block design. Two seeds were sown in each hole at planting and thinned to one plant per hole 10 days after germination. There was no supplemental fertilization, and irrigation was applied twice a month during the plant growth period.

### 2.3. Rhizosphere and Root Sampling

Plants were harvested at the filling stage, which is the stage at which fresh-eating maize is typically harvested. Rhizosphere samples and root samples from each accession were harvested as follows: In each plot, the roots of five independent maize plants were harvested, together with attached rhizosphere soil (loosely attached soils were removed by shaking). Roots with firmly attached rhizosphere soil were pooled to form one biological replicate and quickly stored on dry ice. There were five biological replicates of each accession and density treatment for a total of 20 root + rhizosphere soil samples. After sample collection, roots from each biological replicate were placed in a 2-L Erlenmeyer flask with 500 mL sterile ddH_2_O, and the rhizosphere soil was removed by washing and collected by centrifugation [22]. After the removal of the rhizosphere soil, the remaining roots were washed twice with the same amount of sterile ddH_2_O and used as root samples. After the rhizosphere soil and root samples had been separated, there were a total of 20 rhizosphere soil samples and 20 root samples.

### 2.4. DNA Extraction, Library Construction, and Sequencing

Total DNA was extracted from the rhizosphere soil and root samples according to the instructions of the PowerSoil DNA extraction kit (Mobio Laboratories, Carlsbad, CA, USA). The concentration of the resulting DNA was measured and quantified with a NanoDrop 1000 spectrophotometer (Thermo Scientific, Waltham, MA, USA). The specific primers 799F (5’-AACMGGATTAGATACCCKG-3´), 1193R (5´-ACGTCATCCCCACCTTCC-3´), and 1392R (5´-ACGGGCGGTGTGTRC-3´), targeted to the V5–V7 regions of 16S rDNA, were used for PCR. PCR was performed using phusion high-fidelity PCR master mix with GC buffer (NEB, Ipswich, MA, USA) and the reaction system and amplification program described [23,24,25]. The PCR products were detected by agarose gel electrophoresis, and the target DNA band was purified with the EasyPure Quick Gel Extraction Kit (Transgen Biotech, Beijing, China). The purified DNA was submitted to Majorbio Bio-pharm Technology Co., Ltd., (Shanghai, China) for paired-end library construction and sequencing using the Illumina MiSeq PE300 platform (Illumina, San Diego, CA, USA).

### 2.5. Bioinformatics Analysis

The raw Illumina sequencing data were analyzed using the QIIME2 software platform on a Linux operating system [26]. The paired-end sequences were merged using the vsearch plugin [27], and barcodes and linker sequences were removed with q2-demux and q2-cutadapt trim-pairs [28]. The merged reads were filtered and subjected to quality control and denoising with the quality-filter and deblur plugins in QIIME2 (using a Quality Score > 25) [29]. Sequences were clustered at a 97% similarity threshold determined by the q2-feature-classifier and annotated using the SILVA database. The diversity plugin was used to calculate the alpha diversity, weight-unifrac distance matrix, and Bray–Curtis distance matrix of the bacterial communities. R software (version: 4.0) and the Vegan R package (version: 2.5.6) were used to perform ANOSIM (analysis of similarity) (R core team, 2015). The *R* value and *p* value of the ANOSIM were calculated by permutation testing with permutational anova, and the number of permutations was set to 999. STAMP software (version: 2.1.3) was used with Welch’s *t* test and Benjamini–Hochberg FDR correction to identify microbial taxa whose abundance differed significantly between two treatments [30]. The LEfSe (Linear discriminant analysis effect size) (http://huttenhower.sph.harvard.edu/galaxy/root/index, accessed on 15 November 2022) program was also used with the Kruskal–Wallis rank sum test (alpha of 0.05 and threshold value of 4.8) to test for significant differences in abundance between microbial groups. Bacterial co-occurrence networks in the rhizosphere and roots of maize under different treatments were constructed based on Spearman correlation matrices using the psych R package (version: 1.8.4) [31]. Node connectivity, cumulative degree distribution, and average path length were also analyzed with psych (version: 1.8.4). Correlations with *p* < 0.05 and r > 0.7 were considered to be statistically significant. Network nodes represented genera, and edges connecting these nodes indicated high and significant correlations between genera. Network images were generated using the interactive Gephi platform (Version 9.2, http://gephi.github.io/, accessed on 15 November 2022). The functional prediction of nutrient cycling was carried out using FAPROTAXS software [32].

## 3. Results

### 3.1. Bacterial Community Composition in the Rhizosphere and Roots of Maize

Two fresh-eating maize accessions that are widely cultivated in Fujian Province and show contrasting shoot architecture were selected for this study. L6855 has a compact shoot architecture relative to L27 and is more tolerant of high-density planting (Appendix A). These two accessions were used to study the influence of genotype and planting density on root and rhizosphere microbiota. A field experiment was performed with two planting densities, and roots and rhizosphere soil were sampled at the filling stage for the investigation of their microbial communities. Total DNA was extracted separately from the rhizosphere and root compartments, and the V5–V7 regions of 16S rDNA were amplified using PCR, purified, and subjected to high-throughput sequencing.

The raw sequencing data were analyzed with QIIME2, showing 97% similarity with the number of clustered OTU. Bacterial richness and diversity were significantly higher in the rhizosphere than in the roots but were not significantly affected by planting density (Appendix A). The rarefaction curves showed that the sequencing depth was sufficient to cover most of the bacteria in the root and rhizosphere compartments (Appendix A). After OTU classification and annotation, the analysis of bacterial composition showed that the dominant rhizosphere bacterial phyla were Proteobacteria, Actinobacter, Firmicutes, and Chloroflexi, which together accounted for up to ~95% of all bacteria in the rhizosphere (Figure 1A). Proteobacteria and Actinobacteria were the dominant bacterial phyla in the roots, accounting for up to ~80% of all bacteria. PCoA analysis showed that the rhizosphere bacterial community was distinct from the root bacterial community, as they were clearly separated along PCoA1 (which accounted for 29% of the variation) (Figure 1B). ANOSIM analysis also showed that the bacterial community in the rhizosphere differed significantly from that in the roots (*R* = 0.706; *p* < 0.001). Taken together, these results suggest that an ecological niche is a determining factor that shapes the bacterial communities of the rhizosphere and root compartments in maize.

### 3.2. Microbes Specifically Selected by Maize Roots

To investigate the microbes specifically hosted by maize roots, we identified bacterial taxa that were enriched in the roots compared with the rhizosphere (Figure 1C). Further analysis with STAMP showed that *Bacillus*, *Terrabacter*, *Bryobacter*, *Mesorrhizobium*, *Rhodanobacter*, and *Chujaibacter* were all significantly enriched in the rhizosphere compartment, whereas *Rhizobium*, *Sphingobium*, and *Ralstonia* were highly and significantly enriched in the root compartment, suggesting that they may be specifically selected by maize (Figure 1D). Bacterial co-occurrence networks also differed between the rhizosphere and the roots. Hub microbes in the rhizosphere were dominated by Proteobacteria, whereas those in the roots were dominated by Actinobacteria (Figure 1E). The specific hub microbes in the rhizosphere were *Bradyrhizobium*, *Rhodanobacter*, *Devosia*, *Tumebacillus*, *Massilia*, *Chujaibacter*, *Bryobacter*, and *Intrasporangium*, whereas those in the roots were *Terrabacter*, *Jatrophihabitans*, *Phycicoccus*, *Devosia*, *Nocardioides*, *Cellulomonas*, and *Acidothermus*. Taken together, these results demonstrate that maize roots are colonized by a subset of microbial taxa from the soil.

### 3.3. The Maize Genotypic Traitsthe Structure and Composition of Root-Associated Microbiota

Because previous studies have suggested that a host plant genotype has a substantial effect on the structure and composition of the root microbiota in other plants [33], we investigated the effects of the maize genotype on the root and rhizosphere microbiota. PCoA analysis showed that the rhizosphere bacterial composition of the two maize genotypes could mostly be separated by PCoA2 (accounting for 18.11% of the variation), suggesting that genotypes indeed have an effect on the structure of the rhizosphere bacterial community (Figure 2A). Further analysis with NMDS and ANOSM confirmed that the bacterial community in the L27 rhizosphere differed significantly from that in M6855 (*R* = 0.414; *p* = 0.003) (Figure 2B). Further analysis showed that the relative abundance of some microbes differed significantly between the rhizospheres of L27 and M6855. For example, the relative abundance (RA) of *Actinospica*, *Enterobacter*, and *Rhizobium* was significantly higher in M6855 than in G27, whereas the RA of *Phycicoccus*, *Intrasporangium*, *Pseudolabrys*, *Sinomonas*, *Rhodanobacter*, and *Nitrosospira* was significantly higher in L27 than in M6855 (Figure 2C).

The bacterial community of L27 roots could also be clearly separated from that of M6855 roots by PCoA1 (accounting for 17.55% of the variation), suggesting that the maize genotype also influenced the bacterial community of the root-associated compartment (Figure 3A). NMDS and ANOSIM analyses also confirmed that the bacterial community in G27 roots differed significantly from that in L6855 roots (*R* = 0.32; *p* = 0.001) (Figure 3B). STAMP analysis showed that *Bacillus*, *FCPS473*, and unknown bacteria in the alpha cluster were significantly more abundant in L27 roots than in M6855 roots (Figure 3C). Taken together, our results suggested that the maize genotype had various influences on the bacterial communities in the rhizosphere and roots of maize.

### 3.4. The Influence of Planting Density on Bacterial Communities in the Rhizosphere and Roots of Maize

PCoA analysis showed that the rhizosphere bacterial communities of the two planting densities could be separated by PcoA1 (accounting for 19.76% of the variation) (Figure 4A). NMDS and ANOSIM confirmed that planting density had a significant effect on the rhizosphere bacterial community (*R* = 0.61; *p* < 0.001), and a scatter dot plot showed the effects of planting density on rhizosphere bacteria composition (Figure 4B,C). High-density planting was associated with increased RA of *Arthrobacter*, *Achromobacter*, *Kaistia*, *Pseudomonas*, *Bosea*, *Geodermatophilus*, and *FCP473* in the rhizosphere and decreased RA of *Granulicella*, *Massilia*, *Acidibacter*, *Tumebacillus*, *Bordetella*, and *Geodermatophilus* (Figure 4D). Further investigation indicated that the effect of high-density planting was lower on the root-associated bacterial communities than on rhizosphere communities (*R* = 0.088; *p* = 0.048) (Figure 5A,B). LefSe analysis showed that some specific microbial taxa in the roots were sensitive to planting density (Figure 5C,D): the RA of *Microbacterium*, *Klebsiella*, *Achromobacter*, *Parafrigoribacterium*, *Paenibacillus*, and *Lechevalieria* was significantly higher under high-density cultivation, whereas that of *Pantoea*, *Acidocella*, and *Streptacidiphilus* was lower (Figure 5C). Taken together, these results suggest that planting density had a significant effect on the structure and composition of bacterial communities in both the rhizosphere and roots of maize.

### 3.5. Genotype and Planting Density Have a Synergistic Effect on the Bacterial Community

Because the two maize accessions differed in shoot architecture and thus responded differently to the two planting densities (Appendix A), we also analyzed the combined influence of the genotype and planting density on rhizosphere and root-associated microbial communities. The results showed a clear separation of rhizosphere and root samples from the different treatments (Figure 6A,B), and the combined influence of the genotype and planting density explained 32.7% and 24.7% of the variation in bacterial communities from the rhizosphere and roots, respectively (Figure 6C,D). Detailed analyses showed that the genotype explained 11.1% and 9.97% of the variation in the bacterial communities of the rhizosphere and roots, whereas planting density explained 13.6% and 6.94%.

Co-occurrence network analysis was used to investigate the root and rhizosphere bacterial communities under low and high-density planting conditions. The number of edges connecting taxa in the co-occurrence network was clearly greater under high-density planting (903 edges) than under low-density planting (593 edges) in the rhizosphere compartment (Figure 6E,F). By contrast, no clear difference was observed between the co-occurrence networks of the root bacterial communities under high and low-density planting (Figure 6G,H). Taken together, these results suggest that the bacterial community in the root-associated compartment was less sensitive to planting density than that in the rhizosphere.

### 3.6. Functional Shifts in Rhizosphere Bacteria under Different Planting Densities

We next used FAPROTAXS software to investigate functional changes specifically relative to the nutrient cycling in the root and rhizosphere bacterial communities at different planting densities. The abundance of genes related to some nutrient cycling functions showed significant differences between planting densities in the rhizosphere but not in the roots. Bacterial genes related to nitrate reduction, oxidation of sulfur compounds, nitrate respiration, and nitrogen respiration were significantly increased in the rhizosphere under high-density planting, whereas those related to nitrogen fixation were significantly reduced (Figure 7). No significant changes in the functions of the root bacterial community were observed between planting densities. Taken together, our results suggest that the composition and functions of rhizosphere bacterial communities are more sensitive to planting density than those of the roots.

## 4. Discussion

The structure and composition of root-associated microbiota are influenced by soil type, geographic distance, host plant species, and developmental stage [34,35,36]. The genotype of the host plant is the internal determinant of the establishment of root microbiota [37]. The coordinated co-evolution of host plants and their associated microbes is well known, as is the fact that host plants can actively select specific microbes from the soil to colonize their roots [38,39,40]. Here, we demonstrated that maize selected specific microbes from the soil, as the bacterial community composition and structure of the root endocompartment differed significantly from that of the rhizosphere (Figure 1A–D). The relative abundance of *Rhizobium*, *Sphingobium*, and *Ralstonia* was significantly higher in roots than in the rhizosphere, suggesting that these taxa are selected from the rhizosphere by maize. Comparisons between the two maize accessions showed that within the same plant species, the host genotype has an effect on the structure and composition of bacterial communities in both rhizosphere and roots (Figure 2 and Figure 3). In the rhizosphere, L6855 showed a higher abundance of *Enterobacter*, *Rhizobium*, and *Actinospica*, whereas L27 was enriched in *Sinomonas*, *Rhodanobacter*, and *Phycicoccus* (Figure 2C). These results are consistent with previous studies in which the genotype of the host plant influenced the bacterial communities of the rhizosphere and roots [41,42]. In addition, previous studies on maize have shown that maize could synthesize benzoxazinoids and other root exudates such as sugar and jasmonic acid and affect root-associated microbiome [43,44].

Optimal leaf morphology and sufficient sunlight are the basic conditions for high photosynthetic efficiency. Different maize shoot architectures and planting densities are known to cause differences in photosynthetic efficiency [45]. Specifically, as planting density increased, maize leaf photosynthetic efficiency decreased significantly in a shoot architecture-dependent manner [46]. In maize plants with a horizontal leaf type and larger leaf angles, the upper leaves dramatically overshadow the lower leaves, making these genotypes more sensitive to high-density planting [47]. By contrast, in plants with a compact shoot architecture, lower leaves can maintain high rates of photosynthesis even under high-density planting; such genotypes are usually used for high-density planting in modern agricultural systems [7,48]. Carbohydrates derived from photosynthesis are not only used for growth and development but also as an energy source for root- and rhizosphere-associated microbes through rhizodeposition [18]. Root exudates serve both as carbohydrate and energy sources and as signaling molecules to shape rhizosphere microbial communities [23]. In this study, two maize cultivars with distinct shoot architectures (leaf angles) and contrasting responses to high-density planting (Appendix A) were used to study the interaction of the genotype with planting density in the establishment of root-associated bacterial communities. We found that planting density and genotype explained 13.6% and 11.1% of the variation in the rhizosphere bacterial community and 6.94% and 9.97% of the variation in the root bacterial community (Figure 6C,D).

Our results also demonstrated a significant interaction between the genotype and planting density, which explained 32.7% and 24.7% of the variation in the rhizosphere and root bacterial communities, respectively, suggesting that the host plant genotype influences the response of the root-associated bacterial community to planting density. Our results also suggested that high-density planting had a greater influence on the rhizosphere bacterial community than on the root community (Figure 6E,F,H). Although previous studies addressing the influence of high-density planting are rare, there are some studies that have shown that intercropping with other crops causes significant changes in the rhizosphere of maize or changes to bacterial communities in the rhizosphere of intercropping plants or soils [49,50,51,52]. Previous work has suggested that rhizosphere bacteria are controlled mainly by root exudation, whereas root endophytes are controlled not only by root exudates but also by host plant genotype, including plant nutrient status and immunity [53]. Our results are therefore consistent with a scenario in which high-density planting influences leaf photosynthesis, affecting the distribution and release of photoassimilates through root exudation but having minimal effects on host plant immunity. Light availability may also have an indirect effect on plant growth by changing the composition and function of the bacterial community in root-associated compartments [54]. For example, light may increase carbohydrate allocation to mutualist microbes in the roots [55]. The FAPROTAXS analysis of genes associated with mineral nutrient cycling suggested that planting density had a significant influence on the nitrogen cycle and the oxidation of sulfur compounds. In particular, our results suggest that nitrogen fixation was significantly increased under low-density planting, whereas nitrate reduction, nitrate respiration, and nitrogen respiration were enhanced under high-density planting. These results were consistent in that the changing of the bacterial community mostly led to the change of bacterial functions, shifting relative to nutrient cycling [51,56]. Taken together, these results suggest that different planting densities have significant effects on root-associated bacteria and their functions relative to nitrogen cycling. These results provide a theoretical basis for the manipulation of the root microbiome to enhance maize growth under high planting densities.

## 5. Conclusions

In this study, we analyzed two widely cultivated fresh-eating maize accessions that differ in their shoot architecture and tolerance of high-density planting. We documented differences in the microbial community between the roots and rhizosphere, providing evidence that maize roots selected specific components of the rhizosphere microbiota for root colonization. Genotype and planting density interacted to shape the structure and function of the bacterial community, and planting depth had a stronger effect on the bacterial community in the rhizosphere than in the roots. Taken together, our results provide insights into how aboveground plant architecture and density may alter the belowground bacterial community in the root-associated compartments of maize.

## Figures and Tables

**Figure 1 microorganisms-11-02443-f001:**
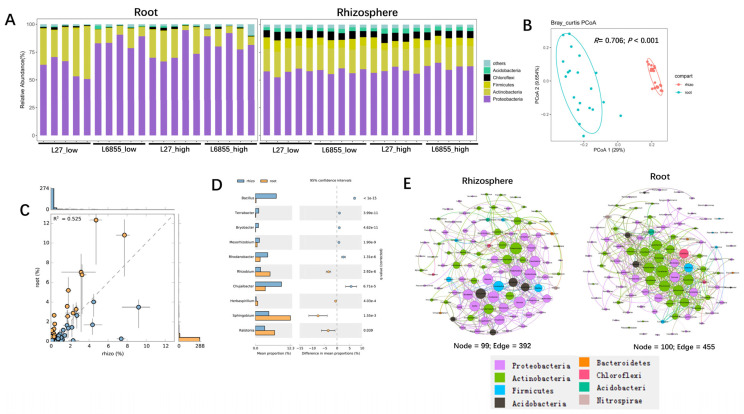
The composition of bacterial communities in the rhizosphere and roots of maize. (**A**) Bacterial community composition at the phylum level in the roots and rhizosphere of maize under low- and high-density cultivation. (**B**) A comparison of bacterial communities between the root and rhizosphere compartments of maize using PCoA and ANOSIM. The *R* and *p* values were calculated using a permutation test with 999 permutations. (**C**) An overall comparison of the difference in abundance of individual bacterial taxa between the root and rhizosphere compartments. (**D**) Bacterial taxa that were significantly enriched in the root or rhizosphere compartment of maize. Corrected *p*-values were calculated using a two-sided Welch’s *t*-test with Benjamini–Hochberg FDR correction. (**E**) A co-occurrence network analysis of bacterial communities in the roots and rhizosphere of maize. Networks were constructed based on a correlation analysis of taxonomic profiles that were significantly (*p* < 0.01; Spearman’s rank correlation test) and highly (Spearman’s *r* > 0.70) correlated. The sizes of the nodes are proportional to the number of connections, and the node colors indicate bacterial phyla.

**Figure 2 microorganisms-11-02443-f002:**
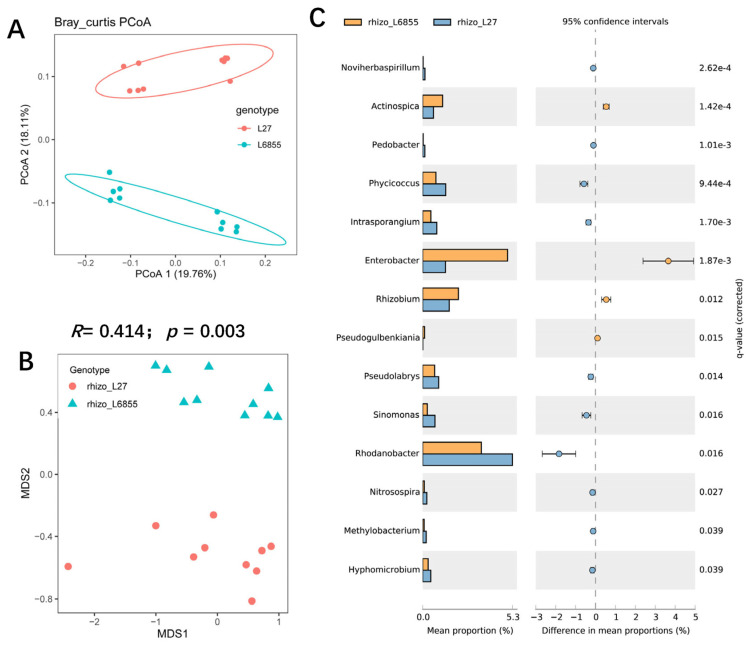
The influence of the maize genotype on the structure and composition of the rhizosphere microbiota. (**A**) A PCoA analysis of the bacterial community in the rhizosphere of two maize genotypes. (**B**) NMDS and ANOSIM were used to compare the rhizosphere bacterial communities of G27 and M6855 maize accessions. (**C**) Bacterial genera whose abundance was enriched in G27 or M6855. Corrected *p*-values were calculated using a two-sided Welch’s *t*-test with Benjamini–Hochberg FDR correction.

**Figure 3 microorganisms-11-02443-f003:**
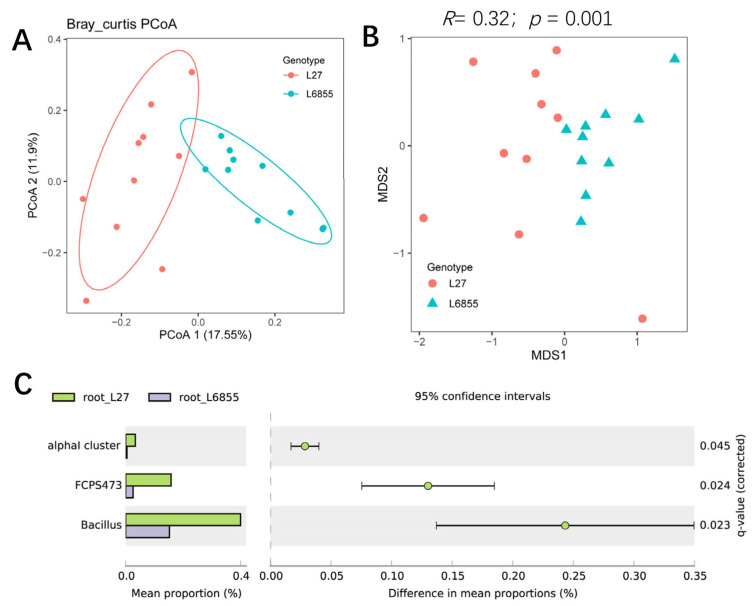
The influence of the maize genotype on the structure and composition of the root-associated microbiota. (**A**) A PCoA analysis of the bacterial community in the root compartment of G27 and M6855. (**B**) NMDS and ANOSIM were used to compare the root bacterial communities between G27 and M6855 maize accessions. (**C**) Bacterial genera whose abundance was enriched in G27 or M6855. Corrected *p*-values were calculated using a two-sided Welch’s *t*-test with Benjamini–Hochberg FDR correction.

**Figure 4 microorganisms-11-02443-f004:**
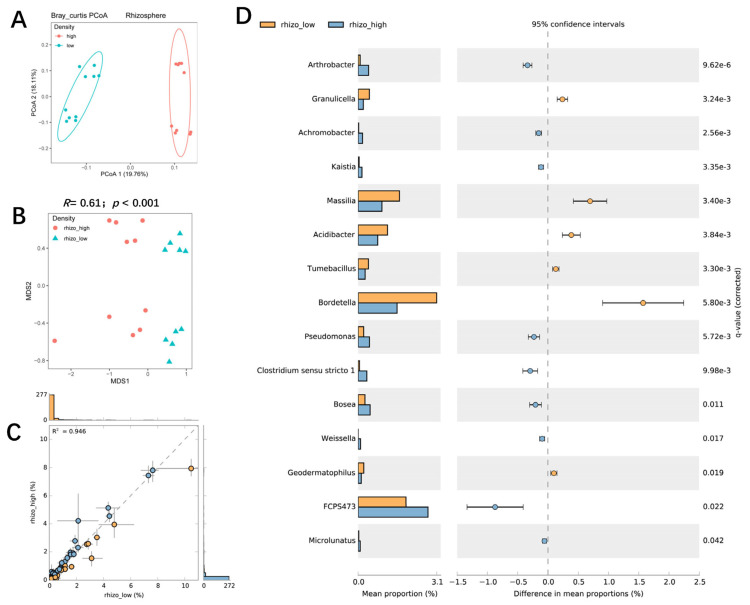
The influence of maize planting density on the structure and composition of the bacterial community in the rhizosphere. (**A**) A PCoA analysis of bacterial communities in the maize rhizosphere under high and low planting densities based on the normalized OTU table. (**B**) A comparison of rhizosphere bacterial communities between high and low planting densities using NMDS and ANOSIM. The R and *p* values were calculated using a permutation test with 999 permutations. (**C**) An overall comparison of the difference in the abundance of individual bacterial taxa between the high and low−density planting treatments. The R^2^ represented the correlation between two samples. Dots next to the dash line indicated little difference between two samples. Dots far away from dashed line indicated obvious differences between two samples. (**D**) Bacterial genera that were significantly enriched between high and low planting densities. Corrected *p*-values were calculated using a two-sided Welch’s *t*-test with Benjamini−Hochberg FDR correction.

**Figure 5 microorganisms-11-02443-f005:**
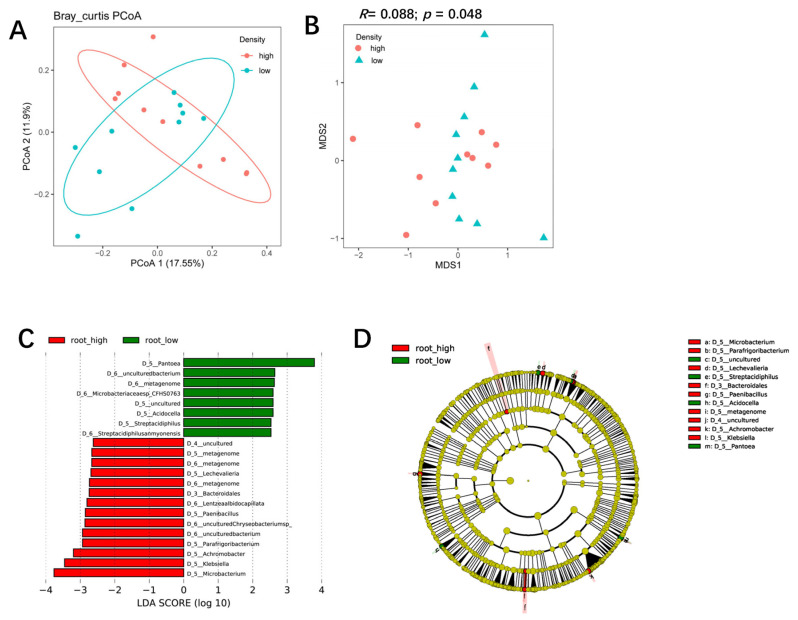
The influence of maize planting density on the structure and composition of the root-associated bacterial community. (**A**) A PCoA analysis of the maize root bacterial community under high and low planting densities based on the normalized OTU table. (**B**) A comparison of root bacterial communities at the community level between high and low planting densities using NMDS and ANOSIM. The R and *p* values were calculated using a permutation test with 999 permutations. (**C**,**D**). Bacterial biomarkers in maize rhizosphere (**C**) and roots (**D**) under high and low planting densities using LefSE analysis. The Kruskal–Wallis rank sum test was used to identify significantly different species within groups at an alpha of 0.05 and a threshold of 2.5.

**Figure 6 microorganisms-11-02443-f006:**
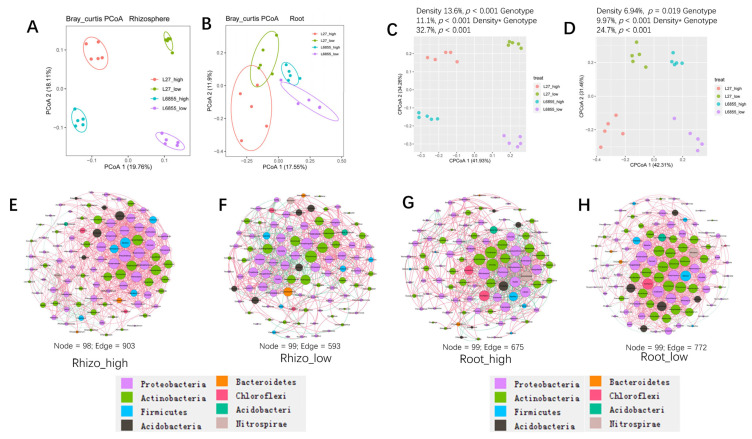
The influence of the density by genotype interaction on the rhizosphere and root bacterial communities. (**A**,**B**) A PCoA analysis of the bacterial community in the rhizosphere (**A**) and roots (**B**) of G27 and M6855 under high- and low-density planting conditions based on the normalized OTU table. (**C**,**D**) The contributions of genotype, planting density, and their interaction to the variation in bacterial community composition in the rhizosphere (**C**) and roots (**D**) of maize. (**E**–**H**) Co-occurrence network analysis of bacteria in the rhizosphere under high-density planting (**E**) and low-density planting (**F**) or in the roots under high-density planting (**G**) and low-density planting (**H**) based on the correlation analysis of taxonomic profiles. Connections are drawn between nodes that were significantly (*p* < 0.01; Spearman’s rank correlation test) and highly (Spearman’s *r* > 0.70) correlated. The sizes of the nodes are proportional to the number of connections, and node colors indicate bacterial phyla.

**Figure 7 microorganisms-11-02443-f007:**
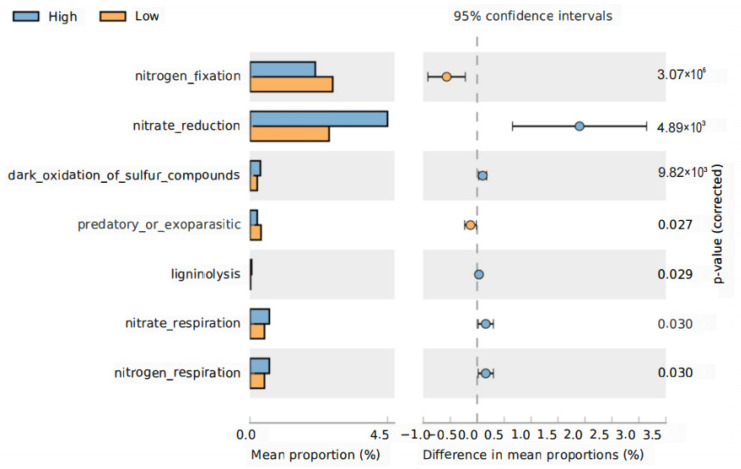
Functional shifts in rhizosphere bacteria under different treatments. Nutrient-cycling genes that showed a significant difference between the high- and low-density treatments were identified using FAPROTAXS and STAMP software. *p*-values were calculated using a two-sided Welch’s *t*-test.

## Data Availability

The 16s rDNA sequencing data have been deposited in the public database of the National Center for Biotechnology Information (NCBI) under project number: PRJNA880986. The source data regarding all the figures have also been submitted in a Appendix A.

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
