# Peer review of "The Influence of the Genotype and Planting Density on the Structure and Composition of Root and Rhizosphere Microbial Communities in Maize"

_microorganisms, 2023, doi:10.3390/microorganisms11102443_

Round 1

Reviewer 1 Report

Comments on the ms. entitled “Influence of maize genotype and planting density on the structure and composition of the root and rhizosphere microbial communities”

Overall

The topic of the manuscript is of high interest to the international readership. The selection of an appropriate cultivation method of two fresh-eating maize cultivars is a highly important issue that will improve the yield and productivity of maize cultivation. The authors used genetic methods to investigate the influence of maize genotype, planting density and their interaction on the composition and function of root and rhizosphere bacterial communities. The results provide insight into the influence of maize genotype and planting density on the interaction between soil microbes and maize roots. The results obtained by the Authors provided new insights into breeding new varieties of corn with high sugar content in grain and laid the groundwork for further research into the mechanism of regulating sugar content in corn grain.

            The paper is well written, the authors formulate the objectives logically and achieve them. I suggest improving the quality of the figures in the final version of the paper, as they illustrate the results obtained well. The text needs minor corrections in the formatting of the text (comments in the manuscript).

Reviewer 2 Report

I'm not convinced at all that high density plantation is essential to maintintain sustainable agriculture. The Aa. are kindly asked to take into consideration other points of view, particularly the need to preserve sustainable cropping systems.

controls should have included (1) low-density of same maize crop genotype said intolerant to low-density planting, (2) low-density planting of another maize crop genotype, different from the one said to be tolerant to low-density planting (3) a high density planting of another maize crop genotype, different from the one said tolerant to high density planting. The description of methods lacks the data on soil composition, particularly because no fertilization was provided. The reference to two selected maize genotypes is not enough: how were they oproduced ? by CRSPR, conventional mutagenesis or else ?

It might be useful, for the improvement of the ms., to extend the discussion on the (strongly expected) influence of maize genotype of microbiome

see above

Reviewer 3 Report

The manuscript titled "Influence of maize genotype and planting density on the structure and composition of the root and rhizosphere microbial communities" investigates the impact of high-density planting and genotype on the root-associated microbiome of maize, revealing significant effects on bacterial communities. The topic holds substantial scientific importance and presents novel insights suitable for publication. However, a major concern regarding the manuscript is the presence of repetitive figures, which necessitates their reorganization. Achieving clarity and coherence in the writing and discussion is contingent upon this figure reorganization. Additionally, it is recommended to provide a clearer and more explicit explanation in the results and discussion sections, highlighting the research's significance on a point-by-point basis. Following a thorough and comprehensive discussion, further consideration will be given to the publication of this study.

Some other suggestions are:

Materials and methods

1.     What are the soil properties of the plots?

2.     What are the suggested spacing for these two varieties? How the spacings of 12/18 cm were chosen?

3.     How were the DNA/samples to be pooled for each replicate?

Results

1.     L174: Explain the values in Welch’s t-test results?

2.     L183: What could be the possible implications for the differences in the enriched microbes? You may put them in the discussion.

3.     L220: The percentage of PC1 + PC2 is too low.

4.     L237: The percentage of PC1 + PC2 is too low.

5.     L240: What could be the possible implications for the differences in the enriched microbes? You may put them in the discussion.

6.     Explain more about figure 4C and also put the legend on the figure.

Round 2

Reviewer 3 Report

The manuscript titled "Impact of Maize Genotype and Planting Density on Root and Rhizosphere Microbial Communities" explores how high-density planting and genotype variations influence the microbiome associated with maize roots. The study uncovers noteworthy effects on bacterial communities, rendering the subject matter of significant scientific value. With some further revisions, the findings are suitable for publication. The suggested revisions are as follows:

1.     A primary concern in this version pertains to the limited proportion of variance explained by PC1+PC2 in the Principal Coordinates Analysis (PCoA). The interpretation of the PCoA results reveals several plausible explanations, such as L204 and L239. If it is stated that "distinct separations were observed," the cumulative variance should ideally exhibit a higher value.

2.     It is advisable to expand discussions regarding the implications of growing density in maize and/or in other plant species. Drawing comparisons with existing studies can enhance the contextualization of the current findings.

3.     On page 309, it would be beneficial to clarify whether the functional analysis solely focused on nutrient cycling. Furthermore, integrating results from previous research studies within the discussion section would underscore the significance of the current discovery.
